# Gastroesophageal reflux disease symptoms and risk of atrial fibrillation in a population-based cohort study (the HUNT study)

**Nikola Drca**[1,2]*, **Malmo Vegard**[3,4], **Jan Pål Loennechen**[3,4], **Imre Janszky**[5,6,7], **Jens W. Horn**[5,8]

**1** Department of Cardiology Karolinska University Hospital, Stockholm, Sweden, **2** Heart and Lung Disease Unit, Department of Medicine Huddinge, Karolinska Institutet, Stockholm, Sweden, **3** Clinic of Circulation and Medical Imaging, Faculty of Medicine, Norwegian University of Science and Technology, Trondheim, Norway, **4** Department of Cardiology, St. Olavs University Hospital, Trondheim, Norway, **5** Department of Public Health and Nursing, Faculty of Medicine and Health Sciences, Norwegian University of Science and Technology (NTNU), Trondheim, Norway, **6** Regional Center for Health Care Improvement, St Olav's Hospital, Trondheim, Norway, **7** Department of Global Public Health, Karolinska Institutet, Stockholm, Sweden, **8** Department of Internal Medicine, Levanger Hospital, Health Trust Nord-Trøndelag, Levanger, Norway

* nikola.drca@ki.se

**Data Availability Statement:** The Trøndelag Health Study (HUNT) has invited persons aged 13–100 years to four surveys between 1984 and 2019. Comprehensive data from more than 140,000

## Abstract

### Aims

Gastroesophageal reflux disease (GERD) may influence the risk of atrial fibrillation (AF). We investigated the association between symptoms of GERD and AF in the Trøndelag Health Study (HUNT).

### Methods

The study cohort comprised 34,120 adult men and women initially free of AF with information on GERD symptoms. Participants were followed from the baseline clinical examination (1 October 2006 to 30 June 2008) to March 31, 2018.

### Results

During a median follow-up of 8.9 years, 1,221 cases of AF were diagnosed. When looking at the whole population, participants with much GERD symptoms did not have an increased risk of AF (HR: 1.01; CI: 95%, 0.82 to 1.24) while participants with little GERD symptoms had a 14% lower risk of AF compared those with no GERD symptoms (HR: 0.86; CI: 95%, 0.76 to 0.97). Among younger participants (<40 years of age), the risk of AF had a trend towards increased risk with increasing symptom load of GERD (little GERD symptoms, HR: 3.09; CI: 95%, 0.74 to 12.94 and much GERD symptoms, HR: 5.40; 95% CI: 0.82 to 35.58). Among older participants (≥65 years of age), we saw a slightly reduced risk of AF in participants with little symptoms (HR: 0.84; CI: 0.72 to 0.97) and no association among those with much GERD symptoms (HR: 1.06; 95% CI: 0.82 to 1.36).

persons having participated at least once and biological material from 78,000 persons are collected. The data are stored in HUNT databank and biological material in HUNT biobank. HUNT Research Centre has permission from the Norwegian Data Inspectorate to store and handle these data. The key identification in the data base is the personal identification number given to all Norwegians at birth or immigration, whilst de-identified data are sent to researchers upon approval of a research protocol by the Regional Ethical Committee and HUNT Research Centre. To protect participants' privacy, HUNT Research Centre aims to limit storage of data outside HUNT databank, and cannot deposit data in open repositories. HUNT databank has precise information on all data exported to different projects and are able to reproduce these on request. There are no restrictions regarding data export given approval of applications to HUNT Research Centre. For more information see: http://www.ntnu.edu/hunt/data.

**Funding:** Center for Innovative Medicine (https://cimed.ki.se/), Region Stockholm, Grant/Award Number: 20190797, Åke Wibergs Stiftelse (https://ake-wiberg.se/), Grant/Award number: M19-0424, Swedish Heart-Lung Foundation (Hjärt-Lungfonden)(https://www.hjart-lungfonden.se/) Grant/Award Number: 20190301, 20200537, and 202110302 and Swedish research Council (Vetenskapsrådet)(https://www.vr.se/) Grant/Award Number: 2019-06102 and 2021-00202 to ND. The funders did not have any role in the study design, data collection and analysis, decision to publish, or preparation of the manuscript.

**Competing interests:** The authors have declared that no competing interests exist.

## Conclusion

We did not find support for a clinically important association between symptoms of GERD and AF across all age groups but for some younger people, GERD might play a role in the development of AF. However, our estimates for this age group were very imprecise and larger studies including younger individuals are warranted.

## Introduction

Atrial fibrillation (AF) is the most common cardiac arrhythmia of clinical significance and the prevalence of AF is estimated to be around 3% among adults [1–3]. Patients with AF have an increased risk of premature death, which can be partly explained by an increased incidence of comorbidity such as hypertension, diabetes mellitus, coronary artery disease, heart failure and stroke [4]. There are multiple risk factors and medical conditions that predispose to the development of AF. The mechanisms behind the onset and maintenance of AF are not fully understood, but left atrial remodeling and ectopic triggers, especially in the pulmonary veins are important contributors. The burden of AF is increasing and identifying modifiable risk factors is pivotal for prevention [5].

Gastroesophageal reflux disease (GERD), defined by troublesome symptoms of heartburn or regurgitation [6] is a common medical condition with an estimated prevalence in adults in Europe of 8.8%-25.9% [7]. Due to the near anatomical location of the esophagus just behind the left atrium and shared esophageal cardiac neural reflex arches, it has been suggested that the development of AF could be associated with the occurrence of GERD [8]. Local atrial inflammation and irritation caused by esophagitis could potentially trigger AF and increased vagal activity could trigger vagal-mediated AF [9]. Esophageal acid stimulation could reduce coronary flow by neural esophageal-cardiac reflex causing angina and eventually ischemic changes in the atrial tissue promoting AF [10]. GERD could be treated with medication (mainly proton pump inhibitors) or surgical intervention with laparoscopic fundoplication [11]. If an association is found between GERD and AF, it could give us a novel opportunity for prevention of AF in selected patients. Previous studies have shown divergent results and had limited possibilities to adjust for potential confounders [12–16]. We therefore investigated the association between self-reported GERD symptoms and the risk of AF in a large population based study with validated AF diagnoses during the follow up and extensive data on potential confounding factors.

## Methods

### Study design and population

All residents in Nord-Trøndelag County who were 20 years and older were invited to participate in four consecutive health survey: the HUNT1 Survey (1984–1986), the HUNT2 Survey (1995–1997), the HUNT3 Survey (2006–2008) and the HUNT4 Survey (2018–2019). The HUNT Surveys include questionnaires that cover a wide range of health-related information, including lifestyle factors, and also clinical measurements and analyses of blood and urine samples [17]. In the present study, we used HUNT3 (participants recruited 1 October 2006 to 30 June 2008) as the baseline. A total of 50 804 residents completed questionnaires and underwent a baseline clinical examination. After excluding participants with missing information on GERD (n = 13,407), potential confounders (n = 2,118) and AF diagnosis (n = 1,159) before

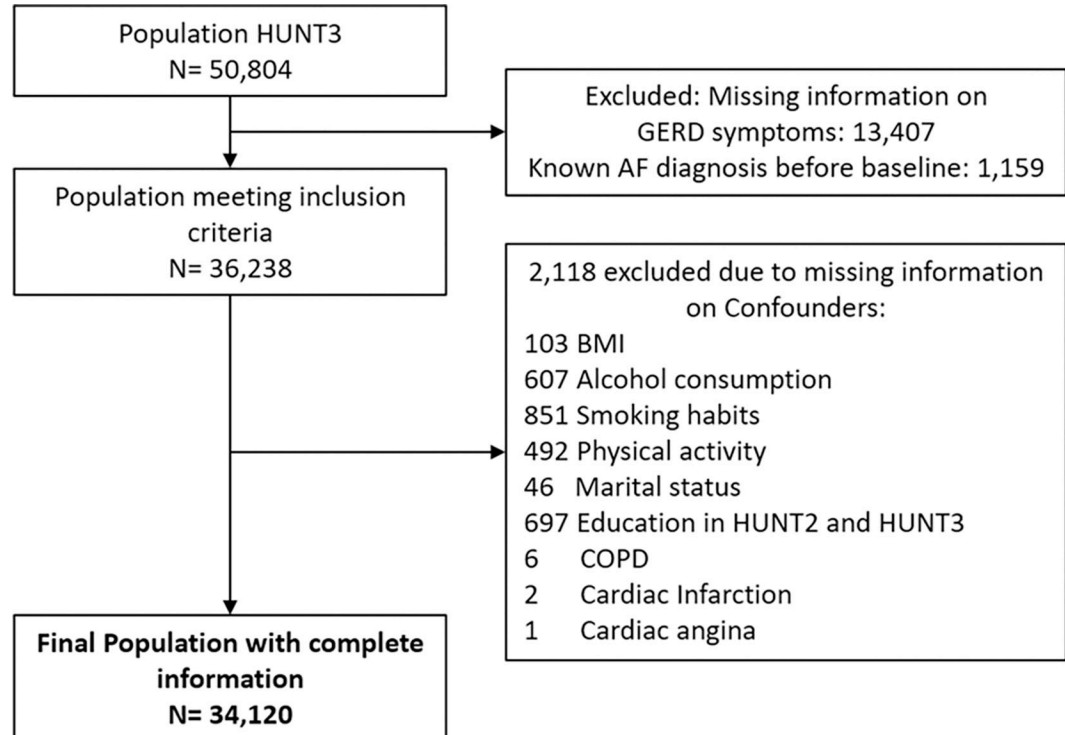

**Fig 1. Final population after excluding participants who did not meet inclusion criteria or with missing information on confounders.** Not mutually exclusive. GERD: Gastroesophageal reflux symptoms; AF: Atrial fibrillation; BMI: Body mass index; COPD: Chronically Obstructive Pulmonary Disease.

baseline, it left us with 34,120 individuals for further analysis (Fig 1). Out of these, 24,550 participants reported on GERD symptoms at HUNT2 (participants recruited 1 August 1995 to 30 June 1997) as well.

The present study was approved by the Regional Committee for Medical and Health Research Ethics Midt-Norway (2016/542 and 2015/2313) and all participants signed a written informed consent before study participation.

## Assessment of GERD (exposure)

In both HUNT2 and HUNT3, information on GERD symptoms was gathered from the participants through a self-administered questionnaire. The participants were asked "To what degree have you had heartburn or acid regurgitation during the previous 12 months?" and were offered three response alternatives "never", "a little" and "much". GERD symptoms in the HUNT 2 have been validated showing that 95% of participants reporting much reflux symptoms experience reflux symptoms at least once a week [18].

## Case ascertainment (outcome)

Incident cases of AF were identified through linkage of participants' Norwegian personal identification number to AF diagnoses (based on ICD-10 code I48) from the two local hospitals in Nord-Trøndelag County from baseline examination date until March 2018. The outcome of AF was defined as either a diagnosis of AF or of atrial flutter because of their close interrelationship. All diagnoses of AF were manually verified by a specialist in cardiology and internal medicine by extracting clinical information and ECG data from the electronic medical records [19].

## Covariates

Information on the participants' smoking status (never, former, occasional and current), physical activity (inactive, active), alcohol use (abstainers, light, moderate and heavy drinkers), education (≤ 9 years, 10–12, > 12 years), marital status (dichotomized as living alone or not) and medical history of common chronic disease such as angina pectoris, myocardial infarction, chronic obstructive pulmonary disease (COPD), diabetes mellitus and hypertension was gathered by a self-administered questionnaire in the HUNT3 Survey. Trained nurses conducted a clinical examination where height and weight were measured wearing light clothes without shoes; height was measured to the nearest 1 cm and weight to the nearest 0.5 kg. BMI was calculated as body weight (in kg) divided by the squared value of height (in meters). Systolic and diastolic blood pressure were measured using an automatic sphygmomanometer based on oscillometry, and the average of the second and third measurement was used in the analysis.

## Statistical analysis

Participants were followed from the baseline clinical examination at HUNT3 to the date of diagnosis of AF, death, emigration out of the county, or March 31, 2018, whichever came first. Continuous variables are presented as mean ± SD while categorical variables are presented as numbers (percentage). We categorized the participants into three groups depending on their self-reported level of GERD symptoms during the last 12 months ("never", "a little" and "much"). Hazard ratios (HR) with 95% confidence intervals (CIs) were estimated using Cox proportional hazards models with age as the underlying time scale. We evaluated the proportional hazards assumption by comparing -ln-ln survival curves and by performing tests on Schoenfeld residuals for each covariate and found no violation of these assumptions. In addition to age at baseline (continuous) and sex, multivariable models were further adjusted for smoking status, alcohol consumption, physical activity level, marital status, education, angina pectoris, myocardial infarction, COPD, diabetes mellitus and hypertension.

Additional analyses were stratified by age (age <40, age ≥ 40 and <65 and, age ≥ 65 years).

All statistical analyses were conducted using Stata 16 for Windows (StataCorp LP, College Station, TX).

## Results

The characteristics of the study population is presented in Table 1. Participants with "much" and "little" symptoms of GERD were older, more often men, had higher BMI, were more often current or former smokers and had more comorbidities such as hypertension, ischemic heart disease, diabetes mellitus and COPD compared to individuals with "never" reflux symptoms. A table comparing the characteristics of in- and excluded participants is presented in the supplementary material. Excluded participants had slightly less cardiovascular diseases and lower prevalence of tertiary education (S1 Table). During a median follow-up of 8.9 years (304,181 person-years), 1,221 cases of AF were diagnosed.

In our main analyses we saw those patients reporting little symptoms of GERD had a multivariable HR of 0.86 (95% CI: 0.76 to 0.97) while patients reporting much had a HR of 1.01 (95% CI: 0.82 to 1.24) compared to patients who never had any GERD symptoms, indicating a U-shaped association (Table 2). Dichotomization of GERD symptoms into just two groups resulted in a HR of 1.08 (95% CI: 0.88 to 1.32) for "Much" GERD symptoms compared to "Never or a little". In a sensitivity analyzes where we excluded all participants who reported cardiac angina, infarction, or COPD symptoms, we observed no relevant changes of the estimates. Additional adjustment for lipids and CRP resulted in no appreciable changes in the estimates (data not shown).

**Table 1. Characteristics of the study population (n = 34,120).**

| | Degree of reported symptoms on Gastroesophageal reflux | | | | | |
|---|---|---|---|---|---|---|
| | Never n = 19,387 | | A little n = 12,314 | | Much n = 2,419 | |
| Female, n (%) | 11,363 | 58.6 | 6,421 | 52.1 | 1,321 | 54.6 |
| Age at Baseline, years (SD) | 51.4 | 15.0 | 54.6 | 14.4 | 54.6 | 14.1 |
| BMI (kg/m$^2$) (SD) | 26.4 | 4.2 | 28.0 | 4.4 | 28.9 | 4.5 |
| Hypertension, n (%) | 6,611 | 34.1 | 5,527 | 44.9 | 1,104 | 45.6 |
| Diabetes mellitus, n (%) | 747 | 3.9 | 656 | 5.3 | 118 | 4.9 |
| C-reactive protein (mg/l) (SD) | 2.31 | 4.85 | 2.95 | 6.00 | 3.37 | 7.03 |
| High-density lipoprotein (mmol/l) (SD) | 1.39 | 0.36 | 1.31 | 0.34 | 1.29 | 0.34 |
| Total cholesterol (mmol/l) (SD) | 5.46 | 1.10 | 5.59 | 1.09 | 5.67 | 1.17 |
| COPD, n (%) | 435 | 2.2 | 527 | 4.3 | 127 | 5.3 |
| Angina pectoris, n (%) | 444 | 2.3 | 503 | 4.1 | 137 | 5.7 |
| Myocardial infarction, n (%) | 426 | 2.2 | 434 | 3.5 | 83 | 3.4 |
| Alcohol consumption | | | | | | |
| Abstainers, n (%) | 3,743 | 19.3 | 2,404 | 19.5 | 524 | 21.7 |
| Light drinkers, n (%) | 11,821 | 61.0 | 7,455 | 60.5 | 1,417 | 58.6 |
| Moderate drinkers, n (%) | 3,548 | 18.3 | 2,293 | 18.6 | 429 | 17.7 |
| Heavy drinkers, n (%) | 275 | 1.4 | 162 | 1.3 | 49 | 2.0 |
| Smoking status | | | | | | |
| Never, n (%) | 9,132 | 47.1 | 4,746 | 38.5 | 818 | 33.8 |
| Former, n (%) | 5,932 | 30.6 | 4,524 | 36.7 | 973 | 40.2 |
| Occasionally, n (%) | 1,300 | 6.7 | 896 | 7.3 | 176 | 7.3 |
| Current, n (%) | 3,023 | 15.6 | 2,148 | 17.4 | 452 | 18.7 |
| Physical inactivity | | | | | | |
| Inactive, n (%) | 3,539 | 18.3 | 2,733 | 22.2 | 647 | 26.8 |
| Active, n (%) | 15,848 | 81.8 | 9,581 | 77.8 | 1,772 | 73.3 |
| Education | | | | | | |
| Lower secondary, n (%) | 3,667 | 18.9 | 3,073 | 25.0 | 674 | 27.9 |
| Upper secondary, n (%) | 9,477 | 48.9 | 6,198 | 50.3 | 1,255 | 51.9 |
| Tertiary, n (%) | 6,243 | 32.2 | 3,043 | 24.7 | 490 | 20.3 |
| Marital Status | | | | | | |
| Single, n (%) | 4,482 | 23.1 | 2,397 | 19.5 | 441 | 18.2 |
| Married, Cohabitant, n (%) | 11,806 | 60.9 | 7,710 | 62.6 | 1,503 | 62.1 |
| Widow, Divorced, Separated, n (%) | 3,099 | 16.0 | 2,207 | 17.9 | 475 | 19.6 |

Values are presented as mean ± standard deviation or number (percentages). BMI body mass index. COPD chronic obstructive lung diseases. n = number.

In the age stratified analyses, among younger participants (<40 years of age), the risk of AF had a trend towards increased risk with increasing symptom load of GERD (little GERD symptoms, HR: 3.09; CI: 95%, 0.74 to 12.94 and much GERD symptoms, HR: 5.40; 95% CI: 0.82 to 35.58). However, due to few events at younger age, the estimates were imprecise with wide Cis. Among older participants (≥65 years of age), we saw a similar U-shaped association as in our main analyses: a slightly reduced risk of AF in participants with little symptoms (HR: 0.84; CI: 0.72 to 0.97) and no association among those with much GERD symptoms (HR: 1.06; 95% CI: 0.82 to 1.36) (Table 3).

To investigate if long-lasting sustained GERD symptoms influence the risk AF, we categorized the population into four groups: Never GERD (No GERD symptoms in HUNT2 and HUNT3)(Reference), Newly Developed GERD (much or little symptoms of GERD in HUNT

**Table 2. Gastroesophageal reflux disease symptoms and risk for atrial fibrillation.**

| GERD Symptoms | n = 34,120 | Events 1,221 | Person years 304,181 | HR* | 95% CI | HR‡ | 95% CI | HR# | 95% CI |
|---|---|---|---|---|---|---|---|---|---|
| Never | 19,387 | 645 | 173,360 | 1 | (Ref) | 1 | (Ref) | 1 | (Ref) |
| A little | 12,314 | 470 | 109,338 | 0.93 | 0.82–1.04 | 0.88 | 0.78–0.99 | 0.86 | 0.76–0.97 |
| Much | 2,419 | 106 | 21,482 | 1.12 | 0.91–1.38 | 1.02 | 0.83–1.26 | 1.01 | 0.82–1.24 |
| Never or a little | 31,701 | 1,115 | 282,699 | 1 | (Ref) | 1 | (Ref) | 1 | (Ref) |
| Much | 2,419 | 106 | 21,482 | 1.16 | 0.95–1.41 | 1.08 | 0.89–1.32 | 1.08 | 0.88–1.32 |

N = number

HR = Hazard Ratio

CI = Confidence interval

*Model 1: Adjusted for age at baseline (continuous) and sex.

‡Model 2: Adjusted as Model 1 and additional for body mass index (continuous), smoking status (never, former, current), alcohol consumption (abstain, light drinkers, moderate drinkers, heavy drinkers), physical activity (inactive, active), education (lower secondary education, upper secondary education, tertiary education) and marital status (unmarried, married/cohabitant, widowed/divorced/separated).

#Model 3: Adjusted as Model 2 and additional blood pressure (dichotomized hypertension: systolic blood pressure>140mmHg or diastolic blood pressure >90 mmHg or use of blood pressure lowering medication), diabetes mellitus (patient indicated to have diabetes mellitus or non-fasting glucose >11.1 mmol/l), chronic obstructive pulmonary disease.

3 but no symptoms in HUNT2), Sustained GERD (much or little symptoms of GERD in both HUNT2 and HUNT 3), Regressed GERD (much or little symptoms of GERD in HUNT2 but no symptoms in HUNT 3). Sustained GERD (HR: 0.90; 95% CI: 0.77 to 1.04) or newly developed GERD (HR: 0.94; 95% CI: 0.79 to 1.11) was not associated with an increased risk for AF. In participants with regression of GERD symptoms, we saw a weak association with a HR of 1.26 (95% CI: 1.01 to 1.56) in the age and sex-adjusted model, which decreased after further adjustment for potential confounders (HR: 1.14; CI: 0.92 to 1.42) (Table 4).

## Discussion

In this large population-based cohort study with validated data on AF diagnosis and robust data on potential confounders, we were able to find a complex association between GERD symptoms and AF. Our main analyses showed a U-shaped association between the risk of AF and the degree of GERD symptoms. In our age stratified analyses, we saw the same U-shaped

**Table 3. Gastroesophageal reflux disease symptoms and risk for atrial fibrillation stratified for 3 age groups.**

| | Age <40 | | | | Age ≥ 40 and <65 | | | | Age ≥ 65 | | | |
|---|---|---|---|---|---|---|---|---|---|---|---|---|
| | n (7,089) | Events (11) | HR | 95% CI | n (19,761) | Events (390) | HR | 95% CI | n (7,270) | Events (820) | HR | 95% CI |
| Never | 4,673 | 3 | 1 | (Ref) | 10,975 | 205 | 1 | (Ref) | 3,739 | 437 | 1 | (Ref) |
| A little | 2,035 | 6 | 3.09 | 0.74–12.94 | 7,296 | 152 | 0.87 | 0.70–1.07 | 2,983 | 312 | 0.84 | 0.72–0.97 |
| Much | 381 | 2 | 5.40 | 0.82–35.58 | 1,490 | 33 | 0.88 | 0.61–1.28 | 548 | 71 | 1.06 | 0.82–1.36 |

n = number; HR = Hazard Ratio; CI = Confidence interval

Adjusted for age at baseline (continuous), sex, body mass index (continuous), smoking status (never, former, current), alcohol consumption (abstain, light drinkers, moderate drinkers, heavy drinkers), physical activity (inactive, active), education (lower secondary education, upper secondary education, tertiary education), marital status (unmarried, married/cohabitant, widowed/divorced/separated), blood pressure (dichotomized hypertension: systolic blood pressure>140mmHg or diastolic blood pressure >90 mmHg or use of blood pressure lowering medication), diabetes mellitus (patient indicated to have diabetes mellitus or non-fasting glucose >11.1 mmol/l) and chronic obstructive pulmonary disease. (Model 3).

P-value for age-GERD interaction < 0.001.

**Table 4. Long lasting gastroesophageal reflux disease symptoms from HUNT2 to HUNT3 and risk for atrial fibrillation.**

| Long Lasting GERD Symptoms | n | Events | Person years | HR* | 95% CI | HR‡ | 95% CI | HR# | 95% CI |
|---|---|---|---|---|---|---|---|---|---|
| Reference. Never GERD symptoms HUNT2 and HUNT3 | 11,811 | 449 | 105,456 | 1 | (Ref) | 1 | (Ref) | 1 | (Ref) |
| Group 1. Newly Developed GERD symptoms HUNT3 | 5,117 | 208 | 45,432 | 1.00 | 0.84–1.17 | 0.95 | 0.81–1.12 | 0.94 | 0.79–1.11 |
| Group 2. Sustained GERD symptoms HUNT2 and HUNT3 | 5,769 | 286 | 50,830 | 1.00 | 0.86–1.16 | 0.91 | 0.78–1.05 | 0.90 | 0.77–1.04 |
| Group 3. Regressed GERD symptoms HUNT2 | 1,853 | 99 | 16,196 | 1.26 | 1.01–1.56 | 1.16 | 0.93–1.45 | 1.14 | 0.92–1.42 |

N = number, HR = Hazard Ratio, CI = Confidence interval

Long Lasting GERD symptoms definition: Reference: Never GERD symptoms in HUNT2 and HUNT3, Group 1: Newly Developed GERD symptoms: Never GERD symptoms in HUNT2 but HUNT3 (little or much), Group 2: Sustained GERD symptoms: HUNT2 and HUNT3 (little or much), Group 3: Regressed GERD symptoms: HUNT2 (little or much) but never in HUNT3.

*Model 1: Adjusted for age at baseline (continuous) and sex.

‡Model 2: Adjusted as Model 1 and additional for body mass index (continuous), smoking status (never, former, current), alcohol consumption (abstain, light drinkers, moderate drinkers, heavy drinkers), physical activity (inactive, active), education (lower secondary education, upper secondary education, tertiary education) and marital status (unmarried, married/cohabitant, widowed/divorced/separated).

#Model 3: Adjusted as Model 2 and additional blood pressure (dichotomized hypertension: systolic blood pressure>140mmHg or diastolic blood pressure >90 mmHg or use of blood pressure lowering medication), diabetes mellitus (patient indicated to have diabetes mellitus or non-fasting glucose >11.1 mmol/l), chronic obstructive pulmonary disease.

association among older individuals. Among younger individuals, we saw a trend towards increasing risk of AF with increasing symptom load of GERD. However, since AF was rare among young participants, the estimates obtained in these analyses were imprecise.

These findings concur with a recent register-based population study that investigated whether objectively determined esophagitis or Barrett's esophagus increases the risk of AF [12]. The investigators found an increased risk of AF among patients younger than 60 years of age who had a HR of 1.55 (95% CI, 1.27 to 1.88) within the first year of diagnosis but the excess risk decreased afterward (HR: 1.14; 95% CI, 1.06 to 1.22). Older patients (age ≥60 years) did not show an increased risk of AF.

A recent meta-analysis summarized prior research on GERD and AF risk not including the study discussed above. The analyses were based on three cohort and one case-control study (total participants: n = 228 305, participants with GERD: n = 80 171) and did not find support for GERD increasing the risk of AF. The summary RR was 1.06 (95% CI, 0.86 to 1.31) [20]. The three cohort studies included in this meta-analysis showed divergent results. Jeffrey S. Kunz and colleagues showed a weak association between GERD and AF after adjusting for several confounders (RR: 1.08; 95% CI, 1.02 to 1.13) [14] and Chin-Chou Huang and associates were able to reproduce these findings with a HR of 1.31 (95% CI, 1.06 to 1.61) [13]. They further saw that patients with GERD symptoms who received proton pump inhibitory (PPI) therapy had the highest risk of AF. This could suggest that patients with the worst symptoms taking PPI therapy also had the highest risk of AF. The third cohort study by Bunch et al. found instead an inverse relationship between the presence of GERD and AF (HR: 0.81; 95% CI, 0.68 to 0.96) after adjusting for other risk factors [15]. In their study, diagnosis of esophagitis seemed to increase the risk for AF (HR: 1.94; 95% CI, 1.35 to 2.78), but after adjusting for age, gender, hypertension, and heart failure the authors reported that the hazard ratio was no longer statistically significant without specifying the point estimate. In an unadjusted survival analysis, they also found a U-shaped association of GERD symptoms and the risk of AF with the highest risk in patients reporting frequent daily symptoms and patients reporting no symptoms. The lowest risk was seen in patients reporting some or weekly GERD symptoms. None of these studies made any age stratified analyses.

A possible explanation for divergent results in studies investigating the association between GERD and AF could be that GERD may have a bigger impact on the risk of AF among younger persons who have a low underlying risk for AF and where vagal nerve stimulation have been found to have a stronger impact on induction of AF [21]. In older individuals, where established risk factors for AF like overweight, hypertension, and age are predominant, and important for AF development it might be difficult to show any eventual additive influence of GERD on the risk of AF.

The observed small risk reduction of AF in older individuals with little GERD symptoms might reflect that treatment of GERD may lead to reduced risk of AF in those individuals. The risk reduction was not observed in older individuals with more severe GERD symptoms.

## Study strengths and limitations

Compared to previous register-based studies, we were able to adjust for several important confounding factors such as BMI, alcohol consumption and physical activity. Our study population was stable and homogeneous. The diagnosis of AF was carefully verified and validated by extracting clinical information and ECG data from the electronic medical records which minimized the risk for misclassification and ensured an almost complete follow-up.

The main limitation of the study is the possibility of exposure misclassification. In our study, GERD was evaluated only based on self-reported symptoms. No gastroscopic examination or pH monitoring was used to assess the presence of esophagitis, reflux or gastric ulcer. Furthermore, we did not have any information on medications such as non-steroidal anti-inflammatory drugs or hormone therapy that could potentially influence the risk. If patients with "much" GERD were treated with anti-reflux medication such as proton pump inhibitors, it could decrease reflux and inflammation and thus possibly prevent AF. There is also a possibility that patients reporting GERD symptoms were seeking medical attention more often and by that increasing their chance of getting an AF diagnosis. We cannot rule out that our results were affected by uncontrolled or residual confounding. For example, we did not have information on different types of food or stress.

## Conclusion

In conclusion, our study did not find support for a strong, clinically important association between symptoms of GERD and AF across all age groups but for some younger people, GERD might play a role in the development of AF. However, our estimates for this age group were very imprecise and larger studies including younger individuals are warranted.

## Supporting information

**S1 Table. Characteristics of in- and excluded study population.**
(DOCX)

## Acknowledgments

The Trøndelag Health Study (HUNT) is a collaboration between HUNT Research Centre (Faculty of Medicine and Health Sciences, Norwegian University of Science and Technology NTNU), Trøndelag County Council, Central Norway Regional Health Authority, and the Norwegian Institute of Public Health. We thank the Department for Research and Development, and clinicians at the Medical Department, Nord-Trøndelag Hospital Trust, Norway, for extracting the data from the patient registers.

## Author Contributions

**Conceptualization:** Nikola Drca.

**Formal analysis:** Jens W. Horn.

**Software:** Jens W. Horn.

**Supervision:** Jan Pål Loennechen, Imre Janszky.

**Validation:** Malmo Vegard.

**Writing – original draft:** Nikola Drca.

**Writing – review & editing:** Nikola Drca, Malmo Vegard, Jan Pål Loennechen, Imre Janszky, Jens W. Horn.

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
