## [Decision Letter · Decision Letter 0]

23 Jan 2024

PONE-D-23-41606Association between Atrial fibrillation and Gastroesophageal reflux disease in a population-based cohort study (the HUNT study)PLOS ONE

Dear Dr. Drca,

Thank you for submitting your manuscript to PLOS ONE. After careful consideration, we feel that it has merit but does not fully meet PLOS ONE’s publication criteria as it currently stands. Therefore, we invite you to submit a revised version of the manuscript that addresses the points raised during the review process.

We look forward to receiving your revised manuscript.

Kind regards,

Dong Keon Yon, MD, FACAAI, FAAAAI

Academic Editor

PLOS ONE

Additional Editor Comments:

Thank you for submitting your manuscript. The reviewers and I believe it is of potential value for our readers. However, the reviewers have raised a number of very important issues, and their excellent comments will need to be adequately addressed in a revision before the acceptability of your manuscript for publication in the Journal can be determined. We cannot guarantee that your revised paper will be chosen for publication; this would be solely based on how satisfactorily you have addressed the reviewer comments.

# Please receive the English editing service.

# Hazard ratios (HR) with 95% confidence intervals (CIs) were estimated using Cox proportional hazards models with age as underlying time scale.

Please cite the statistical guideline (DOI: https://doi.org/10.54724/lc.2023.e8).

# Please add the sentence.

"A two-sided P less than 0.05 considered significance." -> in Method.

# Reference is too short to draw the conclusion.

Reviewers' comments:

Reviewer's Responses to Questions

**Comments to the Author**

1. Is the manuscript technically sound, and do the data support the conclusions?

Reviewer #1: Partly

Reviewer #2: Yes

2. Has the statistical analysis been performed appropriately and rigorously? 

Reviewer #1: No

Reviewer #2: Yes

3. Have the authors made all data underlying the findings in their manuscript fully available?

Reviewer #1: No

Reviewer #2: Yes

4. Is the manuscript presented in an intelligible fashion and written in standard English?

Reviewer #1: Yes

Reviewer #2: Yes

5. Review Comments to the Author

Reviewer #1: This study aimed to investigate the longitudinal association between GERD and AF using data from the HUNT Study.

I have the following comments:

1: It should be clear in the title that the authors investigated the association between GERD (exposure) and AF (outcome). The title (association between AF and GERD) may give the impression that the authors applied AF as an exposure and GERD as an outcome.

2: Lines 57-61: A reference is needed.

3: Line 66: GERD was already spelled out previously.

4: Line 87: > 30% of participants were excluded for having missing data about GERD. The sociodemographic and clinical data should be compared between those who were excluded and those who were included. The potential discrepancy might have affected the representativeness.

5: Line 88: I recommend the authors to reconsider their complete case analysis approach. They may consider imputation or dummy variable analysis.

6: Line 103: ICD-10 code I48 includes atrial flutter. This should be clarified.

7: The authors have provided data about cholesterol levels and CRP in Table 1, yet they did not adjust for these variables in the regression analysis. This should be reconsidered.

8: BP cut-off for hypertension is either 140/90 mmHg per the European guidelines or 130/80 mmHg per the US guidelines. The application of 130/85 mmHg should have a reference.

9: Table 3: The authors do not have to stratify results by age using the value of median. Age results are usually stratified based on clear definitions, older adults versus young and middle-aged adults for example, with a cut-off of 60 or 65 to define older adults.

10: The main limitation of this study is that GERD was self-reported. Was that diagnosis validated in the HUNT Study or any other studies with similar socio-demographic characteristics? This is a very important point that should be further discussed.

11: GERD and AF have so many common risk factors such as age, obesity, smoking, diabetes, dyslipidemia, heavy alcohol consumption, and dyslipidemia. Stratifying the results by these confounders is essential. The authors may consider performing interaction analyses as well.

12: Lines 210-216: This explanation is not based on scientific evidence. Further, participants aged 54 are not older adults. You may consider this explanation after the analysis is stratified by a different age group cut-off.

13: I recommend adding a 3rd GERD category (any) involving those with any GERD symptoms. Given that the authors have no data about GERD endoscopy or medication history, it is so presumptive to accept that the symptom severity suggested by the participants truly reflected their medical condition.

14: Lines 217-221: To test this hypothesis, the authors should revise their data to see whether GERD patients really improved their lifestyle within the following years.

15: Line 236-238: The conclusion gives the impression that there is a kind of association but not so strong, while no association could be attained. I recommend rephrasing this section.

16: GERD is associated with certain foods and drinks, stress and anxiety, pregnancy, NSAIDs, hormonal therapies, and gastric ulcers. These variables were not controlled in this study. The authors should raise this point in their limitation section.

Reviewer #2: Thank you for the opportunity to review this manuscript. This study investigates the association between conscious levels of gastroesophageal reflux symptoms and the risk of atrial fibrillation in individuals aged 20 years and older. The findings reveal a lack of evidence supporting this association and indicate different trends in this association between younger and older participants. The manuscript is well written, and the sample size is notably large. The ethical information and competing interests have been reported.

The reviewer has several comments.

Major comments:

In general, reports of gastroesophageal reflux disease and atrial fibrillation are not uncommon. Therefore, the content of differences compared to previous findings (interaction with age) becomes more important, which reflects the new information of this research field.

Regarding stratified analysis on age grouping, my perspective is that instead of using the median age to stratify, it is preferable to use age intervals that hold clinical significance. The median age only represents the current demographic of the data in wave 3 of the HUNT study. Using age intervals that have clinical significance, such as 20-39 years, 40-64 years, and 65 years and above, can enhance the applicability of the findings to clinical practice. Individuals aged 20-39 are generally considered to have low risks of cardiovascular diseases and atrial fibrillation (as the authors mentioned in line 182, 210), while the elderly population aged 65 and above exhibits significantly elevated risks of them. In contrast, conscious symptoms of gastroesophageal reflux disease are more common in younger individuals, while older individuals usually have fewer conscious symptoms, but more severe conditions and complications (lead to more misclassifications and possibly null association).

The age interaction is a clever entry point. My suggestion is to present more age-related information, such as the proportion of conscious symptoms in different age groups, and to add more content discussing why individuals of different ages may have different findings. I know the authors have already discussed this aspect, but I feel the content is a bit limited. It would be better to add more information, and mention this interaction in the Abstract and Introduction sections.

Moreover, it is preferable to show the interaction effect between age and gastroesophageal reflux disease levels alongside this stratified analysis.

Another suggestion is to use self-reported/ conscious symptoms of gastroesophageal reflux disease to describe exposure in title and other sections. My concern is that this study identified a weak association, which may be due to self-reporting rather than objective measurement (as the authors mentioned in line 228). Using a precise description can accurately convey the findings.

Minor comments:

1. Atrial fibrillation is the outcome of interest, so it is preferable to put it to the back and change the title to: Association between gastroesophageal reflux disease and incident atrial fibrillation…, or use a similar expression of your preference.

2. The expression “(n = 1,221)” in line 106 is unclear. It is preferable to add details and rephrase it as follows: During the follow-up, 1,221 cases of atrial fibrillation (AF) were diagnosed, or use a similar expression, or delete it because you have already mentioned it in line 141 of the Result section.

3. The expression “Degree of reported symptoms…” in Table 1 is not a column name for participants characteristics, so it is preferable to put it on the top of Never/ A little / Much columns. Besides, these numbers in Never/ A little / Much columns have a same column distance, it is preferable to make related numbers (eg. female n 11,363 and % 58.6) close together or merge them, such as 11,363 (58.6%), which is easier to understand. It is better to change the expression “Age at HUNT3” to “Age at baseline”, I think it's easier to understand. The above is my personal suggestion; the authors can choose to modify it based on their own judgment.

4. All continuous variables are presented as mean ± standard deviation in Table 1, which is not appropriate. For continuous variables with a normal distribution, presenting them with a mean and standard deviation is acceptable. However, for variables with a non-normal distribution, such as C-reactive protein, it is more appropriate to display them with an interquartile range, as the standard deviation is not a meaningful statistical measure for variables with non-normal distribution.

6. PLOS authors have the option to publish the peer review history of their article (what does this mean?). If published, this will include your full peer review and any attached files.

Reviewer #1: No

Reviewer #2: No

---

## [Author Response · Author response to Decision Letter 0]

21 Apr 2024

We would like to thank the editor and reviewers for their insightful input which was very helpful in improving the manuscript. Below we provide a point-to-point reply to the comments and suggestions.

Response: We have adjusted the manuscript to PLOS ONE's style requirements to the best of our ability.

Response: It is not possible to share data. These are sensitive patient data, and we are not allowed to share them with third parties due to laws and regulation. 

The Trøndelag Health Study (HUNT) has invited persons aged 13 - 100 years to four surveys between 1984 and 2019. Comprehensive data from more than 140,000 persons having participated at least once and biological material from 78,000 persons are collected. The data are stored in HUNT databank and biological material in HUNT biobank. HUNT Research Centre has permission from the Norwegian Data Inspectorate to store and handle these data. The key identification in the data base is the personal identification number given to all Norwegians at birth or immigration, whilst de-identified data are sent to researchers upon approval of a research protocol by the Regional Ethical Committee and HUNT Research Centre. To protect participants’ privacy, HUNT Research Centre aims to limit storage of data outside HUNT databank, and cannot deposit data in open repositories. HUNT databank has precise information on all data exported to different projects and are able to reproduce these on request. There are no restrictions regarding data export given approval of applications to HUNT Research Centre. For more information see: http://www.ntnu.edu/hunt/data

Response: The Trøndelag Health Study (HUNT) has invited persons aged 13 - 100 years to four surveys between 1984 and 2019. Comprehensive data from more than 140,000 persons having participated at least once and biological material from 78,000 persons are collected. The data are stored in HUNT databank and biological material in HUNT biobank. HUNT Research Centre has permission from the Norwegian Data Inspectorate to store and handle these data. The key identification in the data base is the personal identification number given to all Norwegians at birth or immigration, whilst de-identified data are sent to researchers upon approval of a research protocol by the Regional Ethical Committee and HUNT Research Centre. To protect participants’ privacy, HUNT Research Centre aims to limit storage of data outside HUNT databank, and cannot deposit data in open repositories. HUNT databank has precise information on all data exported to different projects and are able to reproduce these on request. There are no restrictions regarding data export given approval of applications to HUNT Research Centre. For more information see: http://www.ntnu.edu/hunt/data

Additional Editor Comments:

Thank you for submitting your manuscript. The reviewers and I believe it is of potential value for our readers. However, the reviewers have raised a number of very important issues, and their excellent comments will need to be adequately addressed in a revision before the acceptability of your manuscript for publication in the Journal can be determined. We cannot guarantee that your revised paper will be chosen for publication; this would be solely based on how satisfactorily you have addressed the reviewer comments.

# Please receive the English editing service.

Response: We have carefully checked and revised the language of our manuscript

# Hazard ratios (HR) with 95% confidence intervals (CIs) were estimated using Cox proportional hazards models with age as underlying time scale.

Please cite the statistical guideline (DOI: https://doi.org/10.54724/lc.2023.e8).

Response: We have added the citation as suggested. 

# Please add the sentence.

"A two-sided P less than 0.05 considered significance." -> in Method.

Response: Instead of hypothesis testing, we relied on estimation, and accordingly we provided the point estimates together with confidence intervals. Thus, we did not claim statistical significance for our findings. We recognize that opinions differ regarding the concept of statistical significance but we agree with those who strongly argue against its use in typical biomedical research where no immediate decision is required (see for example https://www.nature.com/articles/d41586-019-00857-9 or the statement of the American Statistical Association: Wasserstein RL, Lazar NA. The ASA’s statement on p-values: context, process and purpose. Am Stat. 2016;70(2):129–133.)

# Reference is too short to draw the conclusion.

Response: We were not able to understand this comment.

Responses to Reviewer # 1

Reviewer #1: This study aimed to investigate the longitudinal association between GERD and AF using data from the HUNT Study.

I have the following comments:

1: It should be clear in the title that the authors investigated the association between GERD (exposure) and AF (outcome). The title (association between AF and GERD) may give the impression that the authors applied AF as an exposure and GERD as an outcome.

Response: Response: We have changed the title to “Gastroesophageal reflux disease symptoms and risk for atrial fibrillation in a population-based cohort study (the HUNT Study)”

2: Lines 57-61: A reference is needed.

Response: We have added a reference (5). “Hindricks G, Potpara T, Dagres N, Arbelo E, Bax JJ, Blomstrom-Lundqvist C, et al. 2020 ESC Guidelines for the diagnosis and management of atrial fibrillation developed in collaboration with the European Association for Cardio-Thoracic Surgery (EACTS): The Task Force for the diagnosis and management of atrial fibrillation of the European Society of Cardiology (ESC) Developed with the special contribution of the European Heart Rhythm Association (EHRA) of the ESC. Eur Heart J. 2021;42(5):373-498.”

3: Line 66: GERD was already spelled out previously.

Response: We have changed to just the abbreviation “GERD”.

4: Line 87: > 30% of participants were excluded for having missing data about GERD. The sociodemographic and clinical data should be compared between those who were excluded and those who were included. The potential discrepancy might have affected the representativeness.

Response: We have included a table in the supplementary material showing the characteristics of in- and excluded study populations and a comment in the Results: “A table comparing the characteristics of in- and excluded participants is presented in the supplementary material. Excluded participants had slightly less cardiovascular diseases and a lower prevalence of tertiary education (Supplemental Table 1)”. Also, participants in the HUNT Study received two main packages of questions, referred to as Questionnaire 1 and Questionnaire-2. GERD symptoms were asked in Questionnaire 2. The response rate for GERD reflected closely the overall response rate of its package which makes systematic missingness for GERD-related reasons unlikely.

5: Line 88: I recommend the authors to reconsider their complete case analysis approach. They may consider imputation or dummy variable analysis.

Response: Additional analyses with multiple imputations of missing covariates for the main analyses only changed the estimates marginally. The total number increased from 34,120 to 36,238, number of cases from 1,221 to 1,339. The HR for “Much GERD symptoms” in the fully adjusted model (Model 3) changed from HR 1.01 95% CI 0.82-1.24 to HR 0.99 95% CI, 0.81-1.21. 

These analyses and data are not added or shown in the current version of the manuscript but can be included if so requested.

6: Line 103: ICD-10 code I48 includes atrial flutter. This should be clarified.

Response: We have clarified in the manuscript under “Case ascertainment (Outcome)” (Methods) that I48 includes atrial flutter. “The outcome of AF was defined as either a diagnosis of AF or of atrial flutter because of their close interrelationship”. 

7: The authors have provided data about cholesterol levels and CRP in Table 1, yet they did not adjust for these variables in the regression analysis. This should be reconsidered.

Response: We have originally refrained from adjusting for these factors as they might be on the causal pathway. However, further adjustment with CRP and cholesterol left the results largely unchanged: (See table in Word-file)

Table for reviewers. Gastroesophageal reflux disease symptoms and risk for atrial fibrillation including additionally total cholesterol and CRP

GERD Symptoms n*=34,120

(including n of cases) HR* 95% CI n‡=33,356

(including n of cases) HR‡ 95% CI n#=33,339

(including n of cases) HR# 95% CI

Never 19,387(645) 1 (Ref) 18,972(633) 1 (Ref) 18,962(633) 1 (Ref)

A little 12,314(470) 0.88 0.78-0.99 12,033(459) 0.87 0.77-0.98 12,028(459) 0.87 0.77-0.98

Much 2,419(106) 1.02 0.83-1.26 2,351(100) 0.98 0.79-1.22 2,349(100) 0.97 0.79-1.20

N= number

HR= Hazard Ratio

CI= Confidence interval

* Model 2: Adjusted as Model 1 and additional for body mass index (continuous), smoking status (never, former, current), alcohol consumption (abstain, light drinkers, moderate drinkers, heavy drinkers), physical activity (inactive, active), education (lower secondary education, upper secondary education, tertiary education) and marital status (unmarried, married/cohabitant, widowed/divorced/separated).

‡Sensitivity analysis with total cholesterol additionally included

#Sensitivity analysis with total cholesterol and C-reactive protein additionally included

These data are not shown in the manuscript, but we can certainly include these data if so requested. 

In the manuscript, we have written under “Results”: “Additional adjustment for lipids and CRP resulted in no appreciable changes in the estimates (data not shown)”

8: BP cut-off for hypertension is either 140/90 mmHg per the European guidelines or 130/80 mmHg per the US guidelines. The application of 130/85 mmHg should have a reference.

Response: We have changed the BP cut-off to 140/90 and redone all the analyses which resulted in only minor changes in the estimates. 

9: Table 3: The authors do not have to stratify results by age using the value of median. Age results are usually stratified based on clear definitions, older adults versus young and middle-aged adults for example, with a cut-off of 60 or 65 to define older adults.

Response: We have redone the age-stratified analyses by dividing the population into three age groups as suggested by Reviewer 1 (this comment and comment 12) and by the first comment raised by Reviewer 2: age <40, age ≥ 40 and <65 and, age ≥ 65.

New results are presented in Table 3 and the manuscript.

10: The main limitation of this study is that GERD was self-reported. Was that diagnosis validated in the HUNT Study or any other studies with similar socio-demographic characteristics? This is a very important point that should be further discussed.

Response: We have added information from a validation study performed on participants from the HUNT Study. 

“GERD symptoms in the HUNT 2 have been validated showing that 95% of participants reporting much reflux symptoms experience reflux symptoms at least once a week.”

Reference (18): Nilsson M, Johnsen R, Ye W, Hveem K, Lagergren J. Obesity and estrogen as risk factors for gastroesophageal reflux symptoms. Jama. 2003;290(1):66-72

11: GERD and AF have so many common risk factors such as age, obesity, smoking, diabetes, dyslipidemia, heavy alcohol consumption, and dyslipidemia. Stratifying the results by these confounders is essential. The authors may consider performing interaction analyses as well.

Response: We have adjusted for these variables which is statistically equivalent but more efficient than stratification. Extensive stratification in order to find stronger or weaker effects in certain subgroups, i.e., to find interactions, in the absence of a priori hypotheses, is generally not recommended, see for example: Wittes J. On looking at subgroups. Circulation. 2009 Feb 24;119(7):912-5. doi: 10.1161/CIRCULATIONAHA.108.836601. PMID: 19237669.

However, if so requested, we are willing to consider further stratified analyses where subgroup differences can be expected based on prior data and/or biological mechanisms.

12: Lines 210-216: This explanation is not based on scientific evidence. Further, participants aged 54 are not older adults. You may consider this explanation after the analysis is stratified by a different age group cut-off.

Response: Based on this comment and comment 9 from Reviewer 1 and Comment 1 by Reviewer 2, we have redone the age-stratified analyses by dividing them into three age groups: age <40, age ≥ 40 and <65 and, age ≥ 65 years.

New results are presented in Table 3 and the manuscript.

13: I recommend adding a 3rd GERD category (any) involving those with any GERD symptoms. Given that the authors have no data about GERD endoscopy or medication history, it is so presumptive to accept that the symptom severity suggested by the participants truly reflected their medical condition.

Response: We have added an analysis to our main analysis where we dichotomize symptoms of GERD into just two groups “Never or a little” versus “Much”. GERD symptoms in the HUNT 2 have been validated showing that 95% of participants reporting “Much” reflux symptoms experience reflux symptoms at least once a week while among participants reporting “little” symptoms only 25% had symptoms at least once a week. We believe that the group with “little” GERD symptoms resembles more “never” than “much” GERD symptoms (Nilsson M, Johnsen R, Ye W, Hveem K, Lagergren J. Obesity and estrogen as risk factors for gastroesophageal reflux symptoms. Jama. 2003;290(1):66-72) and therefore we prefer this categorization instead of the suggested one.

14: Lines 217-221: To test this hypothesis, the authors should revise their data to see whether GERD patients really improved their lifestyle within the following years.

Response: We have looked at the changes (see the results at the end of the answer to this comment), but the picture is not very clear, and we have therefore deleted this part on changes explaining the results from the discussion. (See table in the Word-file)

15: Line 236-238: The conclusion gives the impression that there is a kind of association but not so strong

---

## [Editor Report · Decision Letter 1]

15 May 2024

Gastroesophageal reflux disease symptoms and risk of atrial fibrillation in a population-based cohort study (the HUNT Study)

PONE-D-23-41606R1

Dear Dr. Drca,

We’re pleased to inform you that your manuscript has been judged scientifically suitable for publication and will be formally accepted for publication once it meets all outstanding technical requirements.

Kind regards,

Dong Keon Yon, MD, FACAAI, FAAAAI

Academic Editor

PLOS ONE

Additional Editor Comments (optional):

This is an excellent paper.
---

## [Editor Report · Acceptance letter]

20 May 2024

PONE-D-23-41606R1 

PLOS ONE

Dear Dr. Drca, 

I'm pleased to inform you that your manuscript has been deemed suitable for publication in PLOS ONE. Congratulations! Your manuscript is now being handed over to our production team.

Kind regards, 

on behalf of

Dr. Dong Keon Yon 

Academic Editor

PLOS ONE